# Predicting human body composition using a modified adaptive genetic algorithm with a novel selection operator

Xiue Gao[1,2], Wenxue Xie [1,2], Zumin Wang[1], Tianshu Zhang[1,2], Bo Chen[2]*, Ping Wang[3]

**1** College of Information Engineering, Dalian University, Dalian, Liaoning, China, **2** College of Information Engineering, Lingnan Normal University, Zhanjiang, Guangdong, China, **3** Beijing Kangping technology co. LTD, Beijing, China

* chenbo20040607@126.com

## Abstract

### Background

Changes to human body composition reflect changes in health status to some extent. It has been recognized that these changes occur earlier than diseases. This means that a reasonable prediction of body composition helps to improve model users' health. To overcome the low accuracy and poor adaptability of existing human body composition prediction models and obtain higher efficiency, we proposed a novel method for predicting human body composition which uses a modified adaptive genetic algorithm (MAGA).

### Methods

Firstly, since there are many parameters for a human body composition model, and these parameters are associated, we designed a new parameter selection approach by combining the improved RReliefF method with the mRMR method. Following this, selected parameters were used to establish a model that fits body composition. Secondly, in order to accurately calculate the weight of parameters in this model, we proposed a solution which used a modified adaptive genetic algorithm, taking advantage of both roulette and optimum reservation strategies. This solution also has an improved selection operator. Thirdly, taking the percentage of body fat (PBF) as an example of body composition, we conducted experiments to compare performance between our algorithm and other algorithms.

### Results

Through our simulations, we demonstrated that the adaptability of the proposed model is 0.9921, the mean relative error is 0.05%, the mean square error is 1.3 and the correlation coefficient is 0.982. When compared with the indexes of other models, our model has the highest adaptability and the smallest error. Additionally, the suggested model, which has a training time of 28.58s and a running time of 2.84s, is faster than some models.

**Data Availability Statement:** All relevant data are within the paper and its Supporting Information files.

**Funding:** This work is supported by the Competitive Allocation of Special Funds for Science and Technology Innovation Strategy in Guangdong Province of China (No. 2018A06001) by BC, and the Equipment Pre-research Foundation Project (NO.61400010303) by BC. PW mainly contributed to the data collection and software design of manuscripts and is a paid employee of Beijing Kangping technology co. LTD, which did not provide support in the form of salaries for authors [XEG, WXX, ZMW, TSZ, BC, PW] and did not have any additional role in the study design, data collection and analysis, decision to publish, or preparation of the manuscript. The specific roles of these authors are articulated in the 'author contributions' section.

**Competing interests:** Beijing Kangping technology co. LTD did not provide support in the form of salaries for authors XEG, WXX, ZMW, TSZ, BC, PW. The company has no competing interests with us and this commercial affiliation does not alter our adherence to PLOS ONE policies on sharing data and materials.

## Conclusion

The PBF prediction model established by MAGA has high accuracy, stronger generalization ability and higher efficiency, which could provide a new method for human composition prediction.

## Introduction

Human bodies are composed of muscle, fat, inorganic salt, water and other components, each of which make up a percentage of the total mass of the human body. To a certain extent, changes in human body composition reflect physiological or pathological changes. Therefore, using body composition analysis in clinical practice may be beneficial for preventing disease as well as for treatment and rehabilitation [1]. However, many factors affect body composition. These include age, gender, height, weight, race, ethnicity, medical history, dietary intake, exercise status and so on [2]. Equally, prediction accuracy of body composition tends to be determined by the selection of the method. It is therefore clear that a reliable prediction method for human body composition which has high prediction accuracy, strong adaptability and better efficiency is necessary. To date, human body composition prediction methods have included both linear regression methods and intelligent prediction methods, from which many valuable results have been obtained.

Many researchers working with the linear regression prediction method for human body composition have utilized a bioelectrical impedance measurement (BIA), using gender, age, weight, height and bioelectrical impedance as independent variables for a regression prediction equation of human body composition. This equation is able to estimate fat-mass (FM), fat-free mass (FFM), body fat (BF), total body water (TBW) and so on. The commonly used equations for body composition prediction are those of Wang [3], Deurenberg [4] and Lukaski [5]. Chinese scholars have carried out similar work and proposed several human body composition prediction equations for differing groups of people (such as those of different ages, sex, race and so on). Zhang et al. [6] proposed a PBF prediction equation which made use of the skinfold thickness of female children with a bone age of between six and eight. However, the accuracy of these algorithms is not high. By using a magnetic resonance imaging method, Liu et al. [7] established a regression equation for predicting the visceral and subcutaneous fat of both male and female participants. However, this method is difficult to carry out correctly and is not cost effective. Yang et al. [8] proposed a prediction equation for total water, fat-free lean tissue and adipose tissue based on the BIA method, and measured its correlation using the isotope dilution method. Wang et al. [9] established a prediction equation–also based on BIA–for total water content and fat-free body weight which demonstrates high accuracy rates. These methods predict the model according to a regression equation. However, once the regression equation is determined, it becomes difficult to adjust it according to the situation, which means adaptability is poor. Moreover, the regression equation is established based on a certain range of statistical data, meaning errors tend to be significant.

A range of intelligent prediction methods for human body composition have been proposed by scholars worldwide. Kupusinac et al. [10] suggested an artificial neural network-based measurement method which can train many obtained sample sequence sets to establish an FM prediction model. However, training is slow and results do not always converge. Ferenci et al. [11] compared three differing approaches: linear regression, feed-forward neural networks and support vector machines. Simulation results indicated that support vector

machines predict human body fat percentage slightly better than the other two approaches, meaning it has higher accuracy. Lu et al. [12] described an intelligent BF prediction method based on a 3D-shape. Here, a null shape descriptor was selected using a baseline regression model. This then uses visual cues as morphological priori to improve the prediction baseline, resulting in an improved prediction accuracy of BF. Shao et al. [13] proposed a BF prediction strategy which combines any two of the following four methods: multivariate regression (MR); artificial neural networks (ANN); multivariate adaptive regression splines (MARS); support vector regression (SVR). Their results demonstrated that a combined prediction model is better than an individual model. Zhu et al. [14] suggested a human visceral fat area (VFA) prediction model based on wavelet neural networks. When compared to regression models, this model demonstrated significantly improved prediction. Liu et al. [15] suggested an SVM-based bioelectrical resistance anti-human body fat measurement method to improve their model's generalization ability. However, this is only suitable for a small sample. Chen et al. [16] developed a body component prediction model using Akaike information criterion (AIC)-selected characteristic parameters and then used improved entropy to solve the model. This model's accuracy was improved; however, the algorithm is very complex. The common feature of these models is that they make predictions according to the learning mechanism of the intelligent algorithm, meaning the accuracy of the algorithm is significantly improved. However, their disadvantages are that algorithms are influenced by the sample and may converge prematurely, meaning adaptability is poor.

As discussed above, while linear regression is simple, accuracy is low. In contrast, intelligent prediction algorithm is complex, yet has a high accuracy rate. However, as with linear regression, the existing intelligent prediction approach is influenced by samples and adaptability is poor. Therefore, to obtain a better performance prediction model, we explored the intelligent algorithms of the genetic algorithm series. Genetic algorithms are used for optimization problems in many fields [17], although they have the disadvantages of being easily trapped in local optimization and poor optimization efficiency. To obtain better optimization results in diverse domains, genetic algorithms have been improved by many scholars. Sangaiah et al. [18] proposed a hybrid Taguchi-genetic learning Aagorithm (HTGLA), in which the Taguchi method is inserted between crossover and mutation operations. They found that the algorithm improved prediction accuracy. Rahmani Hosseinabadi et al. [19] described an extended genetic algorithm which reached better solutions in terms of computational times and objective values. The fast genetic algorithm suggested by Khanduzia et al. [20] has higher accuracy and faster running speed, and is the combination of a genetic algorithm and a fast branch and cut method. Additionally, adaptive genetic algorithms (AGAs) were developed to improve the ability to find the optimal solution. The AGA can be adjusted by adaptive genetic parameters to improve an algorithm's adaptability, thereby improving both convergence speed and accuracy. However, since AGAs still tend to achieve partial optimization, various modified adaptive genetic algorithms have been proposed.

Ravindran et al. [21] described an improved adaptive genetic algorithm (IAGA), in which they employed a new scaling technique to avoid premature convergence. In addition, they applied an adaptive crossover and mutation techniques to mask the concept and increase population diversity. Results demonstrate that this algorithm is better at finding global optimal solutions. Seong et al. [22] proposed an enhanced adaptive genetic algorithm which combines an adaptive genetic algorithm (AGA) with conventional invasive weeds optimization (IWO) technology. This combination improved search capability and convergence speed. Yan et al. [23] suggested an improved sparse adaptive genetic algorithm in which the parameters of adaptive crossover operator and the mutation operator were improved to achieve global

optimization. Results showed that this algorithm has a higher accuracy, although its calculation is less efficient.

Zhang et al. [24] were able to improve the chaotic adaptive genetic algorithm. Based on the adaptive crossover operator and the mutation operator, chaotic sequences were introduced to initialize the population, while elite retention and mixed sorting operations were utilized to solve precocity. Simulation results suggested that this algorithm has a stronger global search ability, faster convergence speed and greater robustness. In the improved adaptive genetic algorithm described by Fu et al. [25], a restart strategy was used to solve the algorithm's premature convergence problem, while a greedy strategy was employed to improve optimal solution searching speed. These authors also utilized an adaptive crossover operator and a mutation operator to increase the algorithm's adaptability. Results demonstrated that this algorithm has a higher success rate and a faster solution solving speed than others. Yang et al. [26] suggested a solution for improving an adaptive genetic algorithm based on retention policy. In this model, individuals with a higher adaptability demonstrated a lower probability of crossover and mutation operator, while those with poor adaptability demonstrated adaptive crossover and mutation operator. This solution was found to have a faster convergence speed as well as to be more stable.

Following the above research, we found that modified adaptive genetic algorithms not only solve the problem of poor adaptability for these models, but also prevent falling into local optimum. Surprisingly, they also have stronger optimization capability and high convergence efficiency, although no attempts have been made to apply this to the prediction of human body composition. With the aim of obtaining higher prediction accuracy, stronger generalization ability and faster convergence speed, we developed a modified adaptive genetic algorithm for improving the prediction of human body composition. This algorithm improves the selection operator, accounting for the retention problem of the characteristic parameter (individual) with high adaptability in an initial evolution as well as the degradation of the algorithm in late evolution. More importantly, it can be used to find the optimal weight set for the prediction model, combining the advantages of both the roulette strategy and the optimal retention policies.

It is noteworthy that the fitting model should be built before solving the unknown weight of the prediction model and that the construction of the fitting model also has an important impact on prediction accuracy. Furthermore, its construction depends on the selection of the human characteristic parameters. Therefore, an improved human body composition feature selection algorithm was proposed in this paper. The related background information is discussed in the section 'Feature selection and establishment of the body composition fitting model'.

There are three novel contributions in this paper. The suggested feature selection algorithm was firstly used to select the correct feature parameters to build a fitting model. Secondly, we utilized the proposed modified adaptive genetic algorithm to solve the unknown weight set of the model. Finally, a series of simulations were designed to verify the performance of the proposed algorithms. The main details of these contributions are:

1. We propose an improved RReliefF algorithm which combines sample distance metric with sample morphometry and develop a sample similarity distance model which combines human numerical parameter sample distance and sample morphological distance.

2. Based on the improved human body composition feature selection algorithm which combines RReliefF and mRMR, the detailed flow of the algorithm's selected feature is designed to obtain the preferred feature set of human body composition.

3. We propose an improved selection operator which combines both roulette selection and optimal retention strategies. This combined strategy retains the most adaptive individuals while removing the least suitable individuals, ensuring a diversity of individual choices.

4. A flow chart of this human body composition prediction algorithm based on a modified adaptive genetic algorithm is developed, and the weight-solving problem of the human body component prediction model is solved.

5. We designed algorithm simulations to analyse how to choose algorithm parameters, algorithm adaptability and efficiency as well as the proposed algorithm's prediction accuracy of the body composition model.

## Materials and methods

### Ethics statement

The study is consistent with the Helsinki Declaration. Dalian University granted ethical approval to carry out the study within its facilities. The body composition data that was tested in this study, such as height, weight and fat percentage and so on, were not ethically sensitive. We did not test people, but instead collected basic body composition data on the universal INBODY measurement device. The body composition tester (the universal INBODY measurement device) will not cause any harm to the human body. All participants were informed of the purposes of the study and the risks associated with the procedures. Written informed consent was obtained from all participants before the study commenced.

### Feature selection and establishment of the body composition fitting model

Selecting the correct characteristic parameters related to body composition is a key part of the body composition fitting model. In this case, the eight-segment impedance model of the trunk subdivision [27] was used to obtain bioelectrical impedance parameters $R_1 \sim R_8$ of the human body. In addition to considering the eight-segment impedance values, this should also include other physiological parameters that affect body composition such as gender ($S$), age ($A$), height ($H$), weight ($W$), ethnicity ($N$) and other factors. Therefore, basic characteristic parameters such as $R_1 \sim R_8$, $S$, $A$, $H$, $W$ and $R$ are grouped as first characteristic parameters, while the square of the impedance value of each segment in addition to reciprocal and multiple products of every pair are grouped as second characteristic parameters: $R_i^2$, $1/R_i$, $R_iR_j (1 \leq i \leq 8, 1 \leq j \leq 8)$. Combining both first and second characteristic parameters resulted in a candidate feature parameter set of human body composition: $R_1 \sim R_8$, $S$, $A$, $H$, $W$, $N$, $R$, $R_i^2$, $1/R_i$, $R_iR_j (1 \leq i \leq 8, 1 \leq j \leq 8)$.

It is clear that there are many candidate feature parameters whose relationships can be correlation, nonlinearity or irrelevance. Therefore, it is necessary to design a functional feature parameter selection algorithm to remove both uncorrelated and redundant feature parameters. Commonly used feature selection algorithms are often divided into two types: filter and wrapper algorithms. While the initial group has a high computational efficiency, it fails to fully consider redundancy between features. In contrast, the latter group is good at identifying key features, but its calculation speed is slow and it cannot cope well with high-dimensional data sets.

Therefore, in order to obtain a better selection effect, these two types of algorithms–or similar algorithms–are often combined. Zhang et al. [28] proposed the filter-wrapper hybrid feature subset selection algorithm, also known as the maximum Spearman minimum covariance

cuckoo search (MSMCCS). These authors adjusted both correlation and redundancy weights based on the MSMC algorithm and then used both the weighted combination and the cross-mutation strategy in the improved cuckoo search algorithm to obtain the optimal feature sub-set. This algorithm has a higher efficiency and strong classification accuracy. Chen et al. [29] suggested a human physiological feature selection algorithm based on filtering and improving clustering. A feature filtering method based on the Hilbert Schmidt-dependent criterion is utilized to eliminate irrelevant features, while the improved chameleon clustering method removes redundant features, efficiently removing uncorrelated and redundant features. Hu et al. [30] proposed a filtering plus encapsulation hybrid feature selection algorithm which utilizes the partial mutual information method to filter out most unrelated and redundant features while utilizing the FA-based wrapper method to further reduce such features. Use of this method can lead fewer parameters yet more effective features. In addition, this model has a high prediction accuracy.

Chhikara et al. [31] described a hybrid filter-wrapper feature selection algorithm based on improved particle swarm optimization. Multiple regression filtering techniques and t-tests were used to optimize the selection of key features. Improved PSO was used to further reduce the number of features, meaning this method has a higher classification accuracy. Solorio-Fernández et al. [32] proposed a new hybrid filter-package clustering method which categorizes features according to the correlation of features at the 'filter' stage. This method searches for the best feature subset by integrating an improved Calinski-Harabasz index at the wrapper stage as well as utilizing simple ordering and an inversed elimination approach. This algorithm has a higher operational efficiency and is suitable for large datasets. Wang et al. [33] proposed a feature selection algorithm which combined both the RReliefF and mRMR algorithms. The ReliefF is used to calculate the weight coefficients of each feature, while the mRMR algorithm has the optimal correlation within categories as well as minimal redundancy between each category. This algorithm can improve classification accuracy.

Based on the above analysis, it can be seen that the combined algorithms perform better. However, due to the particularity of human physiological characteristic parameters, the new combined selection algorithm should be considered. The physiological parameters of the human body include the impedance of each segment. However, samples of the left and right upper limbs as well as the left and right lower limbs are similar. The existing combination algorithm discussed above does not consider these similarities. Therefore, we suggest an improved RReliefF algorithm which takes advantage of the mRMR algorithm while also considering a sample distance measurement as well as a sample morphology measurement. This algorithm is tailored to the selection of human body component characteristics parameters.

**Improved RReliefF algorithm.** Since human physiological parameters are unique, the bodily values of two people with different physiological parameters (such as impedance, height and weight) may be similar. In contrast, these values may differ for two people with similar physiological parameters. As shown in Table 1, the distance metric (Euclidean distance) between sample 1 and sample 2 is smaller than that between sample 1 and sample 3, while PBF values of sample 1 and sample 3 are similar. This means that when using the RReliefF algorithm to calculate the distance between samples, errors will occur with the original distance

**Table 1. The data of sample distance.**

| Sample | Height (cm) | Weight (kg) | PBF (%) |
|---|---|---|---|
| 1 | 175 | 65 | 21.2 |
| 2 | 177 | 62 | 19.7 |
| 3 | 180 | 68 | 20.9 |

metric, which in turn means that it cannot find a more accurate closest neighbor sample. This means that, in our algorithm, when calculating sample data distance, we also considered the sample morphological distance; that is, the combination of the sample distance metric and the sample morphological metric. Therefore, the RReliefF algorithm is further improved for selecting the parameters of human characteristics.

The Euclidean distance between samples is calculated to find the nearest neighbor sample of the sample using the RRelief algorithm. Eq 1.1 is used to calculate Euclidean distance between sample $i$ and sample $j$:

$$d_{ij} = \sqrt{\sum_{k=1}^{m} (x_{ik} - x_{jk})^2} \tag{1.1}$$

where $x_{ik}$ is the $k$-th body physiological parameter value of sample $i$, $x_{jk}$ is the $k$-th physiological parameter value of sample $j$ and $m$ is the total number of the physiological parameters of the human body. Since both the human body characteristic parameters and the Euclidean distance coefficient have different dimensions, the calculation result must be treated with caution. The relative Euclidean distance coefficient is used as a sample data distance coefficient to solve this. Quality indicators are both standardized and normalized:

$$D_{ij} = \frac{d_{ij} - \min(d_{ij})}{\max(d_{ij}) - \min(d_{ij})} \tag{1.2}$$

where $\min(d_{ij})$ is the minimum and $\max(d_{ij})$ is the maximum value of Euclidean distance. The above equation shows that the norm of the value is closer to 0, suggesting that the distance between the two samples is smaller; when the value is closer to 1, the sample distance is larger.

In order to measure the morphological distance of the sample, the Pearson correlation coefficient of the calculated sample is used. This calculation formula is:

$$r_{ij} = \frac{\sum_{k=1}^{m}(x_{ik} - \bar{x}_i)(x_{jk} - \bar{x}_j)}{\sqrt{\sum_{k=1}^{m}(x_{ik} - \bar{x}_i)^2}\sqrt{\sum_{k=1}^{m}(x_{jk} - \bar{x}_j)^2}} \tag{1.3}$$

where $\bar{x}_i, \bar{x}_j$ are the mean values of the human physiological parameters for samples $i$ and $j$, respectively. Sample morphological distances can be indicated using the absolute value of the similarity coefficient; that is, $R_{ij} = |r_{ij}|(R_{ij} \in [0,1])$. In order to ensure that the morphological distance coefficient and relative Euclidean distance coefficient have a synchronized significance, we used:

$$S_{ij} = 1 - R_{ij} \tag{1.4}$$

The closer the value of the morphological distance coefficient to 0, the greater the similarity between samples. If the value is closer to 1, samples are less similar.

Once impact factors of the sample similarity degree had been normalized, the following sample similar distance model was defined to consider numerical distance of the human physiological parameter sample as well as sample morphological distance:

$$C_{ij} = \alpha D_{ij} + \beta S_{ij} \tag{1.5}$$

where $\alpha$ and $\beta$ are the coefficient weights and $\alpha + \beta = 1$ and the range of sample similarity

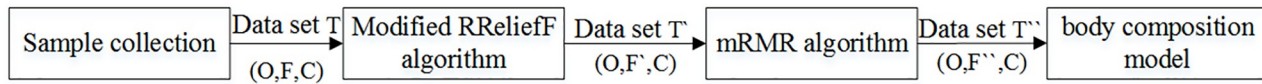

**Fig 1. Selection process of human body physiological characteristic parameters.**

distances $C_{ij}$ is [0,1]. The closer a value is to 0, the more similar the samples. In contrast, the closer a value is to 1, the larger the sample phase difference.

**A body composition feature parameter selection method combining improved RReliefF and mRMR.** The human body feature selection algorithm flow can be seen in Fig 1. Firstly, human physiological information datasets $T = (O, F, C)$ were collected by staff at the physical examination center of one hospital. $O = \{o_1, o_2, \cdots, o_n\}$ represents the training sample data set; $F = \{f_1, f_2, \cdots, f_n\}$ represents the original physiological parameter set; $C = \{c_1, c_2, \cdots, c_n\}$ describes the target category set. Secondly, dataset $T$ is used as the input for the improved RReliefF algorithm. The resulting weighted feature parameter set $F' = \{f_1, f_2, \cdots, f_m\}$ is then used as input dataset $T'$ for the mRMR algorithm. The final optimal feature set $F'' = \{f_1, f_2, \cdots, f_r\}$ was obtained through a combination of improved RReliefF and mRMR methods.

Detailed algorithm flow:

Step 1 Data collection and organization: constructing the training sample dataset $O$, feature parameter set $F$ and target category set $C$; setting up the weighted feature set $F'$ and final target feature set $F''$, all with initial empty values.

Step 2 Sample similarity distance $C_{ij}$ of Eq 1.5 is substituted for sample neighbor distance $d(i, j)$ in the RReliefF algorithm with the aim of improving that algorithm.

Step 3 Training sample data set $O$, feature parameter set $F$ and target category set $C$ are entered into the improved RReliefF algorithm and the weight value of each feature parameter is obtained using formula Eq 1.6, as follows:

$$W[A] = \frac{N_{dC\&dA}[A]}{N_{dC}} - \frac{(N_{dA}[A] - N_{dC\&dA}[A])}{\mathrm{m} - N_{dC}} \tag{1.6}$$

where $W[A]$ is the weight value of the feature parameter; $N_{dC}$ is the weight under various predictive conditions; $N_{dA}[A]$ is the weight under various feature conditions; $N_{dC\&dA}[A]$ is the weight set under various predicted values and feature conditions; and $m$ is the parameter set by the user.

Step 4 Sorting the feature parameter set $F = \{f_1, f_2, \cdots, f_n\}$ by the weighted weight, as calculated in Step 3.

Step 5 Taking characteristic parameters with a weight value greater than a threshold of $\sigma$ into $F'$, resulting in weighted human physiological parameters $F' = \{f_1, f_2, \cdots, f_m\}$.

Step 6 Using obtained weighted human physiological parameter set $F' = \{f_1, f_2, \cdots, f_m\}$ and the target category set $C$ as inputs as well as maximum correlation value $\max D(F', c) = \frac{1}{|F'|} \sum_{f_i \in F'} I(f_i; c)$ in the mRMR algorithm to choose a feature with the highest correlation to the target label to join $F''$.

Step 7 Selecting a new feature parameter from $F'$ to put into $F''$. Assuming that $q-1$ features have been chosen and the target feature set is $F'_{q-1}$, the $q$-th feature is now chosen from the

remaining feature set $\{F'-F'_{q-1}\}$. This satisfies the following equation:

$$\max\left(\sum_{f_i\in F'-F'_{q-1}} I(f_i;c) - \frac{1}{q-1}\sum_{f_j\in F'_{q-1}} I(f_i;f_j)\right) \tag{1.7}$$

Step 8 Repeating Step 7 until the target feature set $F''$ contains $r$ features and classification accuracy $S_r \geq S_{r+1}$. The result is the final feature set.

**Establishment of prediction model for human body composition prediction.** Based on the algorithm described above, a final feature set of $F'' = \{G, A, W, H, R_1R_2, R_2R_3, R_4R_5, R_2, H^2/R_2\}$ is selected, while the fitting model of the obtained human body component PBF can be seen below, in Eq 1.8.

$$f = \omega_1 G + \omega_2 A + \omega_3 W + \omega_4 H + \omega_5 R_1 R_2 + \omega_6 R_2 R_3 + \omega_7 R_4 R_5 + \omega_8 R_2 + \omega_9 H^2/R_2 + b \tag{1.8}$$

where $\omega_1 \sim \omega_9$ are the regression coefficients of the fitted model and $b$ is the model's constant term.

If $X = [x_1, x_2, x_3, x_4, x_5, x_6, x_7, x_8, x_9, 1] = [G, A, W, H, R_1R_2, R_2R_3, R_4R_5, R_2, H^2/R_2, 1]$ then the fitting model equation $f$ is:

$$f(\omega) = \omega X^T \tag{1.9}$$

In the above equation, variable $X$ is known while weight $\omega$ is unknown. This modified adaptive genetic algorithm can be used to solve variables $\omega$.

## Human body composition prediction method based on improved adaptive genetic algorithm

**Roulette selection strategy.** The working principal of the roulette selection strategy starts with a calculation of the fitness and selection probability for each individual, which then form a disc. Secondly, individuals are sorted and numbered in descending order according to selection probability, after which the cumulative probability of the individual can be calculated. Finally, a random number is generated in the interval [1, 0]. If this random number is either less than or equal to the cumulative probability of $i$ individuals or greater than the cumulative probability of $i-1$ individuals, that individual is selected. The probability that any individual is selected is as follows:

$$p(k) = \frac{fit(k)}{\sum_{i=1}^{M} fit(i)} \tag{2.1}$$

where $fit(k)$ is the fitness degree of individual $k$ and $m$ is the population size.

However, there are obvious shortcomings to the roulette selection strategy [34]. In the early stage of the algorithm, individuals with higher fitness are more likely to be selected and copied into the next generation, resulting in a large number of the same individuals in the offspring. A compromised population diversity leads to premature convergence. In the later stage of the algorithm, as most individual differences are not large, each individual's fitness is similar. This means that the roulette selection strategy is likely to lose its ability to choose, and so selection becomes random.

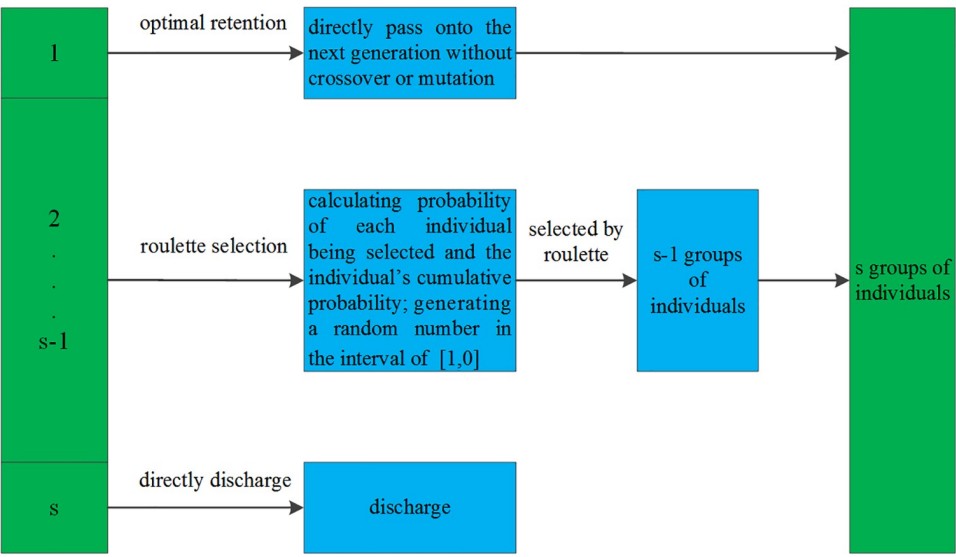

**Fig 2. Improved selection operator strategy.**

**Optimal retention strategy.**   The optimal retention strategy skips crossover and mutation operations on the fittest individuals in the population, and then directly copies the fittest individual to the next generation. This can prevent the existing optimal solution from being destroyed and remove individuals with the worst adaptability from the next generation, ensuring a constant population size. When this strategy is used in a population's evolution and that population reaches a certain scale, the algorithm ensures that the final problem converges to the global optimal solution. However, the algorithm's efficiency is not much improved.

**Improved selection operator combining roulette selection strategy with optimal retention strategy.**   To improve adaptability and diversity of individual selection, we took the advantage of both the roulette selection and the optimal retention strategies. We propose an improved selection operator which combines the two strategies. The details of this operator can be seen in Fig 2. In this operator:

1. All individuals are sorted in their population according to their level of fitness, from large to small, before being equally divided into $s$ groups.

2. The population size is set to $N$. By applying the optimal retention strategy, the $\frac{1}{s}N$ individuals from the group with the best fit are directly passed on to the next generation without crossover or mutation.

3. Individuals from the second group to $s-1$ group were ranked by fitness degree, and Eq 2.1 was used to calculate the probability of each individual being selected. Following this, the individual's cumulative probability was also calculated. The selection operation was then conducted utilizing the roulette strategy.

4. When using the roulette strategy selection, $\frac{s-2}{s}N$ individuals were selected $\frac{s-1}{s}N$ times to obtain $\frac{s-1}{s}N$ individuals.

5. $\frac{1}{s}N$ individuals from the group with the lowest levels of fitness are directly discharged.

Here, differing values of the number of groups *s* produce differing optimal solutions for the population. See Section 3 for a specific analysis. The improved selection operator eliminates the group of individuals with the lowest fitness levels while preserving the highest fitness set. This prevents some possible problems (for example, reversed evolution) while accelerating convergence. When a population is too large and an experimental data sample is overwhelming, the improved algorithm has better efficiency and higher convergence speeds as compared to the optimal retention strategy. Additionally, our algorithm overcomes several disadvantages of the roulette selection strategy. Individuals with greater adaptability do not participate in roulette selection and are instead entered directly to the next generation, ensuring a diverse population. Retention of well-aligned individuals in the later stages ensures the algorithm does not degrade into random selection.

**The flow of the MAGA-based human body composition prediction algorithm.** The principal of MAGA is to solve the unknown variable $\omega$ of the body composition prediction model as follows: the fitness function is set, following which decimal coding and initialized population are utilized. Subsequently, an adaptive selection operator, adaptive crossover operator and adaptive mutation operator are used in order. The optimal solution for the variable $\omega$ is then obtained. This algorithm flow is shown in Fig 3.

## (1) Encoding and initialization

Since the model obtained here is a continuous function and visual representation of the body composition prediction problem, the unknown coefficient representation of this fitting model can be written as $\omega = [\omega_1, \omega_2, \omega_3, \omega_4, \omega_5, \omega_6, \omega_7, \omega_8, \omega_9, b]$. The decimal real number coding method is used. A set of unknown coefficients $\omega = [\omega_1, \omega_2, \omega_3, \omega_4, \omega_5, \omega_6, \omega_7, \omega_8, \omega_9, b]$ represents the individuals in the population. Each individual's gene is a real number, the range of which is set to $[-100,100]$. When the population is initialized, M randomly generates groups of unknown parameters to constitute an initial subpopulation. The initial value of the evolution generation counter is set to 1 and the maximum genetic generation is set to 400.

## (2) Fitness function

Both benefits and drawbacks of body composition prediction can be measured using the degree of similarity between body composition predictions $f_k(\omega)_i$ and real values $F_i$. The closer these two values are, the more accurate the prediction result. Under this condition, the individual $\omega = [\omega_1, \omega_2, \omega_3, \omega_4, \omega_5, \omega_6, \omega_7, \omega_8, \omega_9, b]$ has the highest adaptability degree. We used the mean residuals of predicted and actual values as the objective function of the prediction method:

$$object(k) = \frac{1}{m} \sum_{i=1}^{m} \frac{|f_k(\omega)_i - F_i|}{F_i} \qquad (2.2)$$

where *m* is the size of the training sample, $F_i$ is the actual value of the i-th training sample and $f_k(\omega)_i$ is the predicted value of the *i*-th training sample of the *k*-th individual. Eq 2.2 shows that the smaller the function value, the closer the predicted value is to the real value. In order to represent the optimal solution of the problem in a maximized form, the function is transformed, as can be seen below, while the final fitness function is obtained, as demonstrated in Eq 2.3.

$$fit(k) = \frac{1}{1 + object(k)} \qquad (2.3)$$

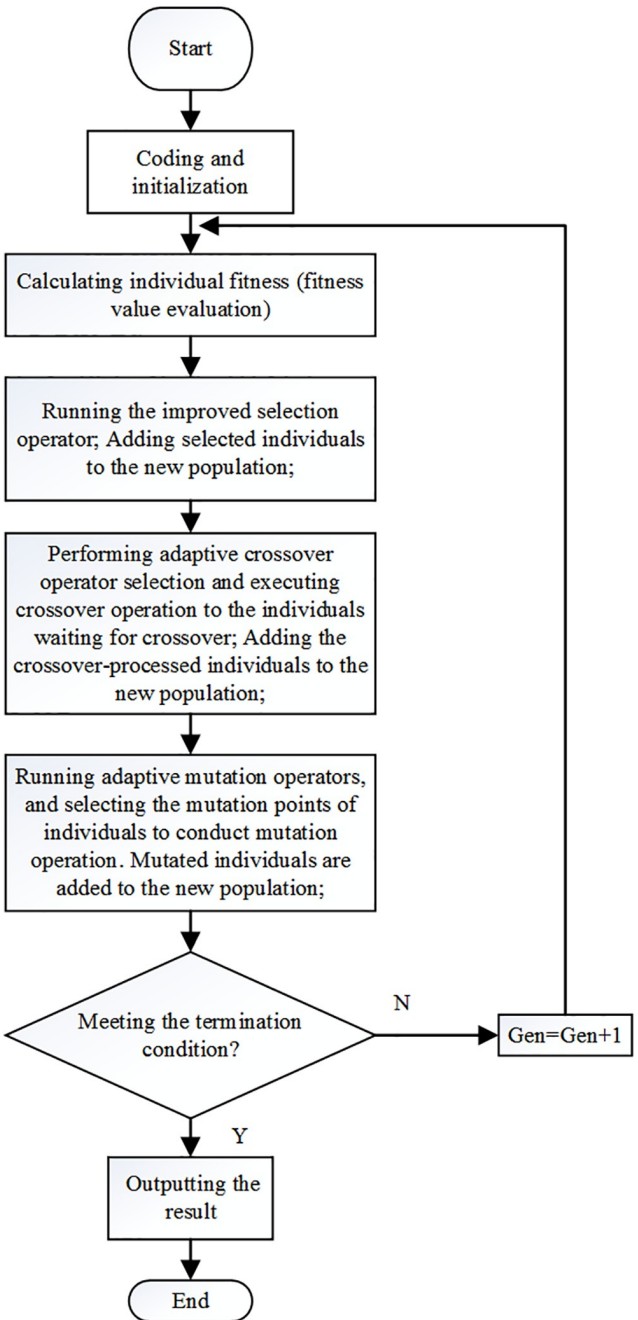

**Fig 3. The flow of the MAGA-based human body composition prediction algorithm.**

where *object*($k$) is the objective function of body composition prediction. This is another form of Eq 2.2, while it is clear that *object*($k$) $\in$ [0, +$\infty$] and that *fit*($k$) $\in$ [0,1].

## (3) Genetic operator

The genetic operator consists of three parts: selection operator, crossover operator and mutation operator. The improved method proposed in this article is used when selecting the

operator execution. The construction of both the adaptive crossover operator and the adaptive mutation operator is the same as that utilized in several existing studies [35, 36, 37].

## (4) Iterative termination condition

When the error between the predicted and actual values is less than or equal to 0.01, or when the number of iterations reaches a maximum value of 400, iteration is terminated, while the optimal individual in the current population becomes the solution output of the problem.

## Results and discussion

### Experimental data

The laboratory care committee of Dalian University granted ethical approval to carry out the study within its facilities. The dataset is the physiological characteristic parameters of 220 healthy Chinese participants, including 96 females and 124 males, who ranged in age from nine to 84 years, in height from 149 to 190cm, and in weight from 38.4 to 121.4kg from one hospital (the Haidian Maternal and Child Health Hospital) in Beijing, China. The parameters are $G$, $A$, $W$, $H$, PBF and $R_1 \sim R_8$, which respectively represent the variables of gender, age, weight, height, percentage of body fat and human body eight-segment impedance. Additionally, $R_1 \sim R_8$ represents the impedance of the human body's left upper limb, upper trunk, right upper limb, left trunk, right trunk, lower trunk, left lower limb and right lower limb respectively. A total of 200 samples were randomly selected as the training set, while the remaining 20 samples were used as test sets. Training set data can be seen in Table 2, while test set data is shown in Table 3. The modified adaptive genetic algorithm proposed in this paper was used

**Table 2. Human physiological characteristics of training samples.**

| NO. | G | A | W(kg) | H(cm) | $R_1(\Omega)$ | $R_2(\Omega)$ | $R_3(\Omega)$ | $R_4(\Omega)$ | $R_5(\Omega)$ | $R_6(\Omega)$ | $R_7(\Omega)$ | $R_8(\Omega)$ | PBF(%) |
|---|---|---|---|---|---|---|---|---|---|---|---|---|---|
| | | | | | | | Human physiological characteristic parameters | | | | | | |
| 1 | M | 28 | 82.0 | 175 | 253.9 | 20.1 | 264.7 | 21 | 21.7 | 26.1 | 212.5 | 212.0 | 27.8 |
| 2 | F | 51 | 68.6 | 154 | 277.5 | 22.9 | 293.2 | 23.5 | 23.9 | 29.4 | 205.4 | 200.8 | 39.1 |
| 3 | M | 23 | 109.5 | 182 | 235.3 | 20.4 | 250.9 | 20.7 | 21.3 | 25.9 | 166.8 | 164.9 | 35.1 |
| 4 | M | 48 | 80.9 | 171 | 225.3 | 21.6 | 236.1 | 22 | 22.6 | 28.1 | 175.9 | 174.9 | 28.0 |
| 5 | F | 74 | 54.9 | 150.5 | 327.9 | 23.1 | 336.5 | 23.5 | 24.2 | 26.3 | 264.8 | 274.6 | 37.7 |
| 6 | F | 26 | 59.2 | 163 | 405.3 | 20.8 | 390.8 | 21.2 | 21.9 | 27.7 | 235.8 | 238.2 | 31.4 |
| 7 | M | 23 | 109.5 | 182 | 232.1 | 21.9 | 247.1 | 22.3 | 23 | 28.7 | 166 | 167.6 | 34.6 |
| 8 | F | 22 | 50.4 | 162.5 | 362.8 | 22.7 | 376.9 | 23.1 | 24.2 | 27.1 | 263 | 258.5 | 24.5 |
| 9 | M | 52 | 92.4 | 180 | 240.3 | 20.6 | 236.7 | 21.2 | 21.9 | 27.6 | 179.9 | 190.6 | 26.2 |
| 10 | F | 40 | 65 | 160 | 327.6 | 21.9 | 332.7 | 22.4 | 23.5 | 24.5 | 241.5 | 236.9 | 35.7 |
| 11 | F | 34 | 42.8 | 155 | 340.9 | 21.1 | 356 | 21.9 | 22.2 | 29.5 | 216.4 | 220.5 | 18.3 |
| 12 | M | 59 | 71 | 169 | 241.8 | 21.5 | 242.6 | 22.1 | 22.6 | 27.8 | 210.5 | 209.4 | 23.6 |
| 13 | F | 57 | 65 | 153 | 264.4 | 22.2 | 266.3 | 22.6 | 23.5 | 29.4 | 219.3 | 222.3 | 37.3 |
| 14 | M | 50 | 89 | 172 | 248.6 | 21.5 | 246 | 23.5 | 20.4 | 27.4 | 199.2 | 207.5 | 34.4 |
| 15 | F | 84 | 63.8 | 151.5 | 276.4 | 23.2 | 291.2 | 23.8 | 25.4 | 33.2 | 194.1 | 199.9 | 38.4 |
| 16 | F | 49 | 65 | 159.5 | 291.1 | 22.3 | 295.5 | 22.8 | 23.6 | 30.1 | 219.3 | 217.2 | 35.3 |
| 17 | M | 61 | 79.3 | 169 | 265.1 | 21.9 | 286.4 | 22.4 | 22.2 | 28.3 | 211.6 | 206.3 | 34.3 |
| 18 | M | 55 | 65.2 | 163 | 279.2 | 21.3 | 296.4 | 23.1 | 23.7 | 30.4 | 241.5 | 243.3 | 30.5 |
| 19 | F | 45 | 49.7 | 152.5 | 367.7 | 21.5 | 377.1 | 22.1 | 21.9 | 25.8 | 261.5 | 267.7 | 32.3 |
| ... | ... | ... | ... | ... | ... | ... | ... | | | ... | ... | ... | ... |
| 200 | M | 22 | 77.5 | 183 | 279.7 | 21.3 | 291.3 | 21.7 | 22.9 | 29.4 | 224 | 231.9 | 20.5 |

**Table 3. Human physiological characteristics of the test samples.**

| NO. | G | A | W(kg) | H(cm) | $R_1(\Omega)$ | $R_2(\Omega)$ | $R_3(\Omega)$ | $R_4(\Omega)$ | $R_5(\Omega)$ | $R_6(\Omega)$ | $R_7(\Omega)$ | $R_8(\Omega)$ | PBF(%) |
|-----|---|---|-------|-------|------|------|------|------|------|------|------|------|--------|
| | | | | | | | Human physiological characteristic parameters | | | | | | |
| 1 | F | 60 | 59.5 | 162 | 307.0 | 23.4 | 297.6 | 23.9 | 24.4 | 30.5 | 261.9 | 217.7 | 31.1 |
| 2 | M | 23 | 49.3 | 158 | 370.5 | 22.4 | 362.1 | 22.7 | 23.5 | 29.3 | 257.4 | 262.9 | 27.8 |
| 3 | M | 28 | 58.5 | 180 | 307.3 | 24.5 | 312.3 | 25.4 | 26.4 | 32.3 | 261.8 | 255.7 | 12.6 |
| 4 | M | 28 | 58.5 | 180 | 312.5 | 25.9 | 322.2 | 26.5 | 27.3 | 34.2 | 259.6 | 252.4 | 13.8 |
| 5 | F | 51 | 56.9 | 161 | 341.1 | 23.6 | 340.6 | 25 | 27.2 | 34.4 | 230.1 | 227.7 | 29.6 |
| 6 | F | 29 | 79.3 | 163 | 278.9 | 26.4 | 278.5 | 26.9 | 27.3 | 33.5 | 200.1 | 184.2 | 40.6 |
| 7 | M | 59 | 77.2 | 176 | 244.8 | 22.2 | 260.4 | 22.5 | 22.8 | 29.1 | 192.7 | 190.7 | 22.9 |
| 8 | M | 45 | 87.8 | 180 | 235.7 | 20.3 | 237 | 20.9 | 21.3 | 26.4 | 221.2 | 215.6 | 23.0 |
| 9 | M | 57 | 75.0 | 163 | 235.4 | 20.6 | 244.9 | 21.2 | 21.5 | 28 | 192.6 | 192.7 | 32.2 |
| 10 | M | 31 | 78.4 | 171 | 216.9 | 19.8 | 222.7 | 20.3 | 20.7 | 25.8 | 167.4 | 160.7 | 21.6 |
| 11 | F | 42. | 55.8 | 152 | 282.8 | 26.6 | 291.5 | 27.4 | 28 | 33.4 | 216.2 | 212.7 | 31.4 |
| 12 | F | 51 | 68.3 | 162 | 307.4 | 20 | 306.3 | 20.7 | 21.3 | 27.2 | 193.7 | 188.8 | 33.9 |
| 13 | F | 54 | 70.5 | 168 | 290.0 | 28.1 | 296 | 28.7 | 29.1 | 35.2 | 208.6 | 203.3 | 30.2 |
| 14 | M | 55 | 60.7 | 165 | 284.7 | 23.3 | 274.9 | 23.8 | 24.3 | 30 | 222.6 | 219.1 | 21.9 |
| 15 | M | 50 | 79.7 | 166 | 241.6 | 26.7 | 241.9 | 27.3 | 27.6 | 33.7 | 193.0 | 199.4 | 30.7 |
| 16 | F | 33 | 50.0 | 156 | 322.0 | 25.3 | 327.5 | 25.9 | 26.7 | 32.1 | 241.9 | 243.1 | 24.7 |
| 17 | M | 43 | 64.2 | 170 | 292.4 | 23.1 | 283.7 | 23.8 | 24.3 | 29.8 | 236.6 | 245.9 | 21.7 |
| 18 | M | 43 | 66.4 | 175 | 303.7 | 21.1 | 320 | 22.3 | 21.9 | 28.1 | 237.3 | 229.1 | 21.1 |
| 19 | M | 51 | 63.5 | 168 | 255.3 | 26.1 | 259.1 | 26.7 | 27.5 | 32.1 | 304.4 | 206.0 | 27.6 |
| 20 | M | 48 | 76.6 | 172 | 264.2 | 22.2 | 272.3 | 23.5 | 22.3 | 29.9 | 205.5 | 203.4 | 27.5 |

for modeling training. The model was implemented and tested in Matlab R2016a using Python 3.5.4 software environment. The mean square error of the PBF prediction model can be seen below:

$$MSE = \frac{1}{n}\sum_{i=1}^{n}\left(f_k(x)_i - \overline{f_k(x)}\right)^2 \tag{3.1}$$

where $n$ is the test set size, $f_k(x)_i$ is the predicted value obtained each time from the model and $\overline{f_k(x)}$ is the mean of the predicted values.

The correlation coefficient is calculated by comparing predicted results with actual values. The calculation method is:

$$r(f_k(x)_i, F_i) = \frac{Cov(f_k(x)_i, F_i)}{\sqrt{Var[f_k(x)_i]Var[F_i]}} \tag{3.2}$$

where $Cov(f_k(x), F_i)$ is the covariance of $f_k(x)_i$ and $F_i$, $Var[f_k(x)_i]$ is the variance of $f_k(x)_i$, $Var[F_i]$ is the variance of $f_i$, $f_k(x)_i$ is the predicted PBF value of the body composition model and $F_i$ is the actual PBF value.

## Experimental results and discussion

Experiments on algorithm parameter selection and model performance comparison are reported in this paper. The choice of algorithm parameters has a crucial impact on the prediction effect, so we firstly explored how to choose the important parameters of the feature selection algorithm and MAGA.

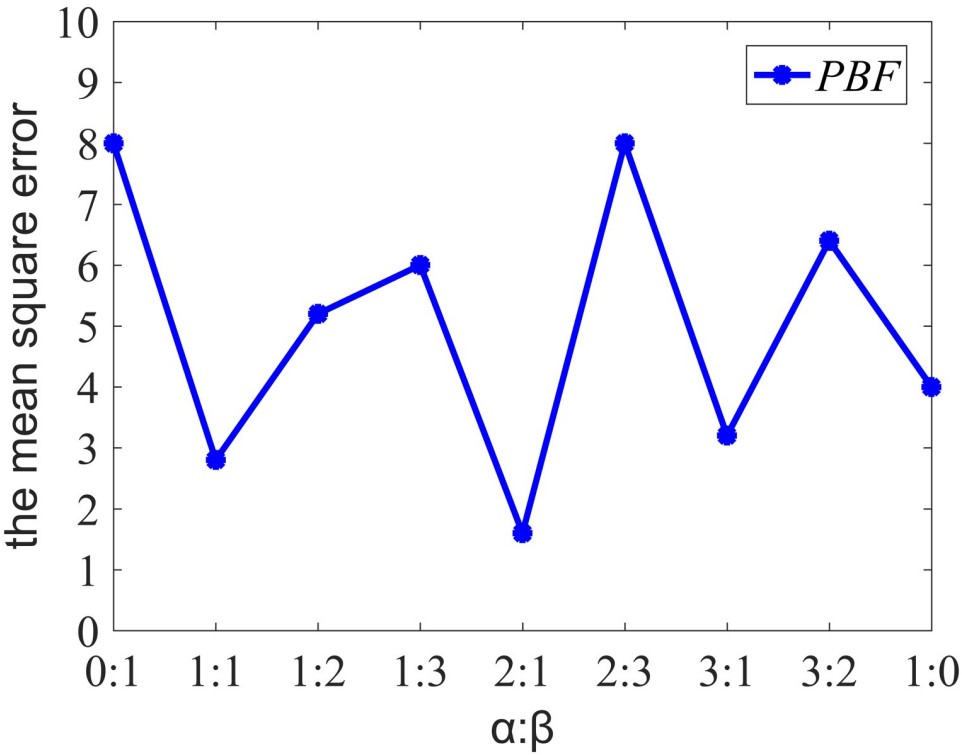

**Fig 4. Mean square error of different ratios of $\alpha$ and $\beta$.**

To explore the influence of different values of key parameters ($\alpha$ and $\beta$) on the prediction accuracy in the feature selection algorithm, the comparison experiment was set. Fig 4 demonstrates how the different ratios of $\alpha$ to $\beta$ affect PBF prediction errors and that the mean square error of the prediction model is the smallest when the ratio of $\alpha$ to $\beta$ is 2: 1. In addition, according to formula 1.5, $\alpha+\beta = 1$. Therefore, the value of $\alpha$ is set to one third and the value of $\beta$ is set to two thirds in the feature selection algorithm, thus obtaining a more satisfying prediction accuracy.

The setting of the parameter $s$ in the MAGA has a great influence on fitness. It can be seen from Fig 5 that when $s$ is 8, the population fitness is the highest. If the value is either too large or too small, it is not conducive to finding the optimal solution for that population. If it is too small, the grouping is also too small while the optimal retention is too large and the population evolution is slow, meaning that better individuals may not be selected. However, if it is too large, the grouping is also too large, and the optimal individual retains too little to prevent the optimal solution being found. Hence, we set the parameter $s$ to 8 for selecting the higher fitness individual.

To examine the performance of the proposed model, we compared the body composition prediction models based on genetic algorithm (GA), adaptive genetic algorithm (AGA), improved adaptive genetic algorithm (IAGA) [35, 36, 37], support vector regression algorithm (SVR) [13, 38], artificial neural network algorithm (ANN) [10], traditional regression method (TRM) and MAGA.

Fig 6 demonstrates the comparative analysis results of the fitness of different models, while Table 4 is a comparison of the fitness and training time of differing models. Since TRM, SVR and ANN algorithms are not involved in the calculation of fitness function, there are no maximum fitness degrees in Table 4. Fig 6 shows that once a population has evolved 110 times, the

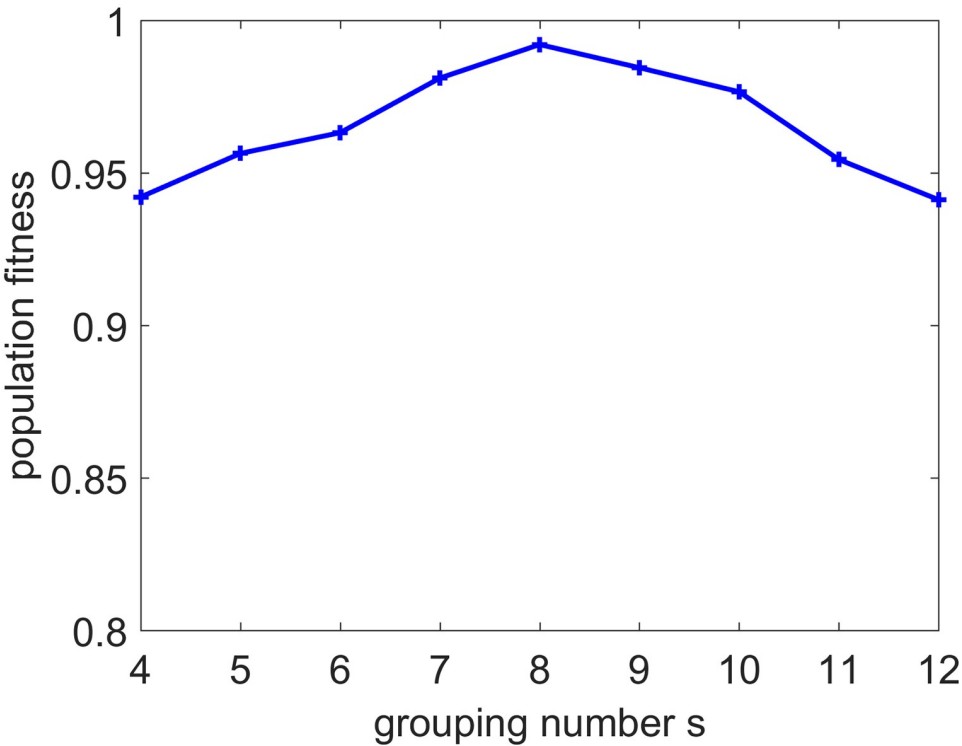

**Fig 5. Relationship between grouping number *s* and population fitness.**

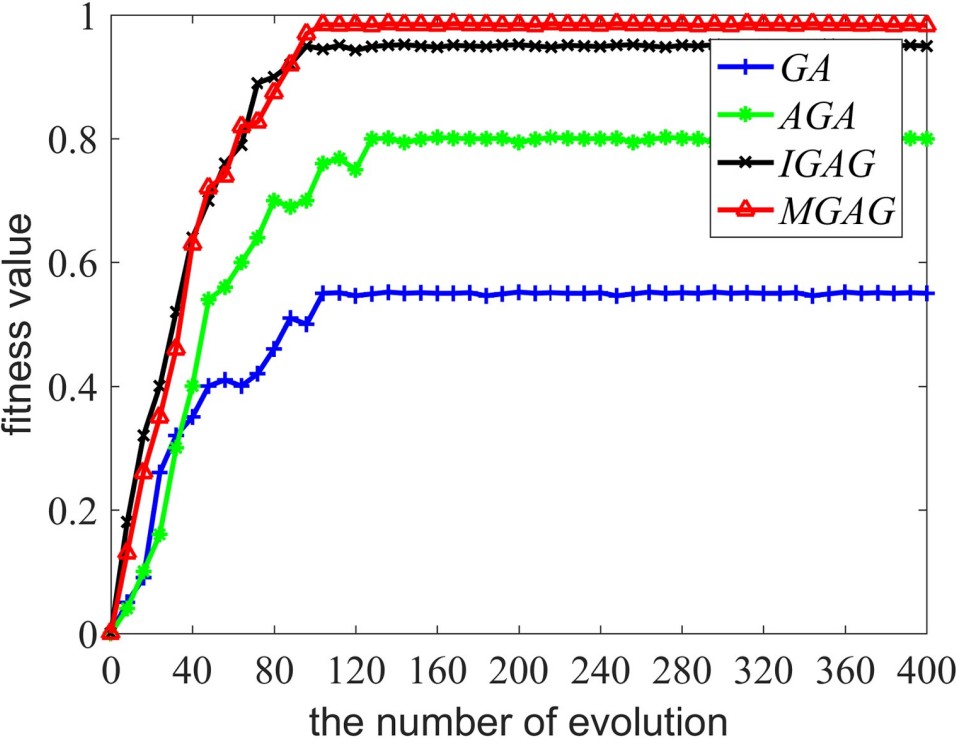

**Fig 6. Comparative analysis of the fitness of different models.** The legend for Fig 6 is "GA, AGA, IAGA, MAGA".

**Table 4. Comparison of the training time for different models.**

| Model | Maximum fitness degree | Training time(s) |
|---|---|---|
| GA | 0.5642 | 26.82 |
| AGA | 0.8074 | 28.23 |
| LAGA | 0.9532 | 28.94 |
| MAGA | 0.9921 | 28.58 |
| SVR | - | 30.67 |
| ANN | - | 29.89 |
| TRM | - | 25.46 |

MAGA model fitness, which is the largest among the compared models, reaches a maximum of 0.9921, meaning the value of the iterative evolutionary fitness no longer increases and the optimal solution is obtained. Furthermore, it is noteworthy that the MAGA model fitness has been increasing during population evolution, but the fitness of other models may occasionally decline, demonstrating a better robustness. With the use of both the adaptive crossover operator and the mutation operator, our improved selection operator is able to protect the optimal solution while ensuring diversity of choice. This means that, when compared to other algorithms, the MAGA improves the model's fitness and robustness. Table 4 shows that, when compared with the models based on IAGA, SVR and ANN, training time for the MAGA model is slightly shortened while there is no significant improvement for the MAGA model compared with the models based on GA, AGA and TRM. Since the premature problem of both GA and AGA is serious and the obtained population is not highly adapted, the models' training time is short. The MAGA's improved selection operator is superior to the selection operator of the IAGA and MAGA has a faster searching optimal solution speed compared with SVR and ANN, meaning that training time is shorter.

Fig 7 demonstrates the comparison of predicted values obtained by different models with their actual values. Fig 8 shows the comparison of relative errors between predicted and actual values obtained by different models, while Table 5 shows a comparison of the indicators of different algorithms.

It can be seen from Fig 7 that, when compared to other models, the predicted value curve generated by the MAGA model has the smallest difference from the actual value curve. As shown in Fig 8, when compared with other models, relative errors between predicted values generated by our MAGA model and actual values are the smallest. Table 5 shows that the mean relative error of the proposed model is 0.05% and the mean square error is 1.3. Additionally, the correlation coefficients between the predicted and true values of the six models are all greater than 0.5 and the P values are all far less than 0.01, which show that they are strongly correlated and the results are of statistical significance. In particular, the correlation coefficient of the proposed model is 0.98 and the P value is 0. In sum, this shows that the predicted value obtained by the proposed model is the closest to the real value and the overall error rate is the lowest of all models, meaning it has the highest prediction accuracy. Moreover, while the running time is slightly lower than the IAGA, SVR and ANN models in the same environment, it is a little more than the AGA and TRM models. However, although the AGA and TRM models have higher operating speeds, their prediction accuracy is very low.

Based on the above results, the MAGA model not only has better robustness and higher adaptability, but also can quickly find the optimal solution. What is more, the accuracy of the prediction model is more desirable. Accordingly, there are significant advantages for prediction of PBF using the suggested algorithm.

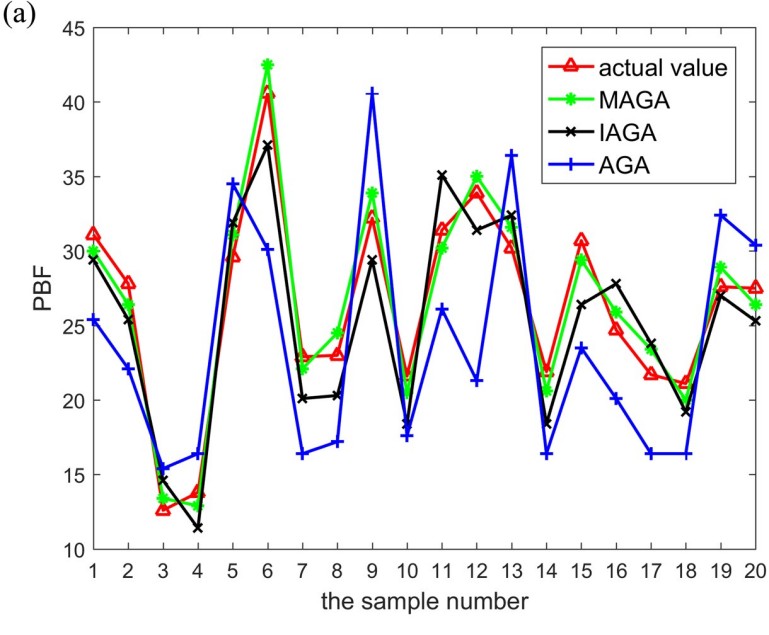

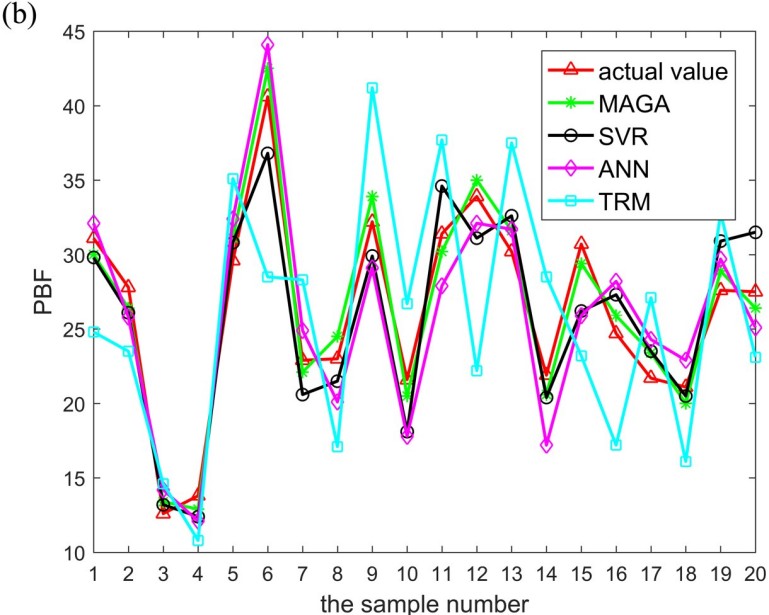

**Fig 7. Comparison of the actual values and predicted values obtained by different models.** The legend for Fig 7-(a) is "actual value, MAGA, IAGA, AGA". The legend for Fig 7-(b) is "actual value, MAGA, SVR, ANN, TRM". (a) Comparison of the actual values and predicted values obtained by the MAGA, IAGA and AGA models. (b) Comparison of the actual values and predicted values obtained by the MAGA, SVR, ANN and TRM models.

## Conclusions

Since existing methods of body composition prediction are impacted by the sample and demonstrate poor algorithm adaptability and low accuracy, we have proposed a new method for predicting body composition. In this method, preferred human body characteristic parameters were firstly obtained using a feature selection algorithm which combines RReliefF and mRMR. The body composition prediction fitting model was established based on this algorithm.

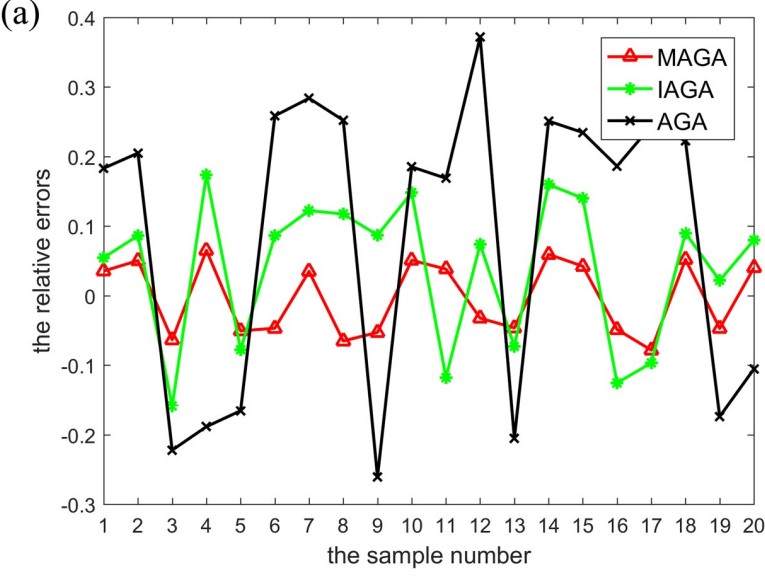

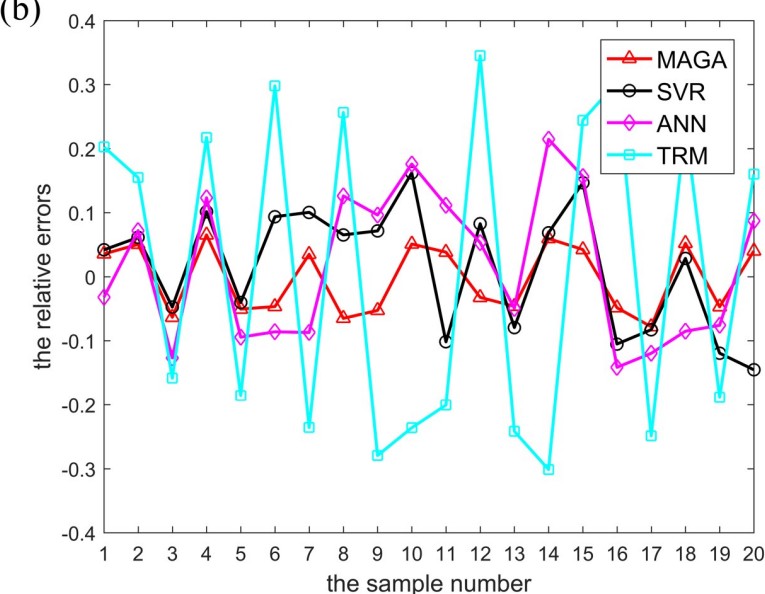

**Fig 8. Comparison of relative errors between actual values and predicted values obtained by different models.** The legend for Fig 8-(a) is "MAGA, IAGA, AGA". The legend for Fig 8-(b) is "MAGA, SVR, ANN, TRM". (a) Comparison of relative errors between the actual values and predicted values obtained by the MAGA, IAGA and AGA models. (b) Comparison of relative errors between the actual values and predicted values obtained by the MAGA, SVR, ANN and TRM models.

Secondly, the roulette selection and optimal retention strategies were used for adaptation. The selection operator of the genetic algorithm was also improved, and this new algorithm was used to solve the unknown weight in the fitting model. Finally, an example sample was used to simulate the proposed method's effectiveness. Simulation results show that the method improved the accuracy of the body composition prediction model, while operation efficiency and model fitness degree were also improved. The advantages of this method may arise from the following aspects: 1) due to the appropriate feature selection algorithm, characteristic

**Table 5. Comparison of the performance of different models.**

| Model | Mean relative error (%) | Mean square error | Correlation coefficient | Running time(s) |
|---|---|---|---|---|
| AGA | 0.2185 | 6.27 | 0.684[a] | 2.66 |
| LAGA | 0.1045 | 2.72 | 0.929[b] | 2.97 |
| MAGA | 0.0500 | 1.30 | 0.982[c] | 2.84 |
| SVR | 0.0873 | 2.56 | 0.927[d] | 3.02 |
| ANN | 0.1058 | 2.85 | 0.920[e] | 2.95 |
| TRM | 0.2348 | 6.73 | 0.596[f] | 2.49 |

[a]. P value (significance level) is 0.001, which is far less than 0.01.

[b]. P value (significance level) is approximately 0, which is far less than 0.01.

[c]. P value (significance level) is approximately 0, which is far less than 0.01.

[d]. P value (significance level) is approximately 0, which is far less than 0.01.

[e]. P value (significance level) is approximately 0, which is far less than 0.01.

[f]. P value (significance level) is 0.006, which is far less than 0.01.

parameters with large correlations and small redundancy were selected; 2) the combination of roulette selection and optimal retention strategies ensured the diversity of selection; 3) the superiority of the proposed modified adaptive genetic algorithm means it can obtain the optimal solution for the problem faster and more accurately. We anticipate that this algorithm could provide a new model reference for human body composition prediction. We will conduct further research in the future based on the following five aspects: 1) we will consider how to more effectively reduce the time complexity of the algorithm; 2) cross-validation will be used to test model performance; 3) the influence of different sample sizes on the prediction effect will be considered; 4) we will explore the predictive effects of the proposed algorithm on more body components such as total body water and body fat; 5) the combination of the two intelligent algorithms will be considered to solve the unknown weights in the body composition fitting model.

## Supporting information

**S1 File. Data set.**
(XLSX)

## Author Contributions

**Conceptualization:** Xiue Gao, Wenxue Xie, Zumin Wang, Tianshu Zhang, Bo Chen.

**Data curation:** Wenxue Xie, Tianshu Zhang, Ping Wang.

**Formal analysis:** Xiue Gao, Wenxue Xie, Zumin Wang, Tianshu Zhang.

**Funding acquisition:** Bo Chen.

**Investigation:** Xiue Gao, Wenxue Xie, Zumin Wang, Bo Chen.

**Methodology:** Xiue Gao, Wenxue Xie, Tianshu Zhang.

**Project administration:** Xiue Gao, Zumin Wang, Bo Chen.

**Resources:** Xiue Gao, Zumin Wang, Bo Chen.

**Software:** Wenxue Xie, Tianshu Zhang, Ping Wang.

**Supervision:** Xiue Gao, Bo Chen.

**Validation:** Xiue Gao, Wenxue Xie.

**Visualization:** Xiue Gao, Wenxue Xie, Zumin Wang.

**Writing – original draft:** Xiue Gao, Wenxue Xie, Zumin Wang, Bo Chen.

**Writing – review & editing:** Xiue Gao, Wenxue Xie.

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
