## [Decision Letter · Decision Letter 0]

20 Dec 2019

PONE-D-19-20446

Predicting human body composition using a modified adaptive genetic algorithm

PLOS ONE

Dear Mr. Chen,

Thank you for submitting your manuscript to PLOS ONE. After careful consideration, we feel that it has merit but does not fully meet PLOS ONE’s publication criteria as it currently stands. Therefore, we invite you to submit a revised version of the manuscript that addresses the points raised during the review process.

We would appreciate receiving your revised manuscript by Feb 02 2020 11:59PM. To enhance the reproducibility of your results, we recommend that if applicable you deposit your laboratory protocols in protocols.io, where a protocol can be assigned its own identifier (DOI) such that it can be cited independently in the future. For instructions see: http://journals.plos.org/plosone/s/submission-guidelines#loc-laboratory-protocols

We look forward to receiving your revised manuscript.

Kind regards,

Le Hoang Son, Ph.D

Academic Editor

PLOS ONE

**J**ournal requirements:

2. Thank you for stating the following in the Competing Interests section:"The authors have declared that no competing interests exist." 

We note that one or more of the authors are employed by a commercial company: Beijing Kangping technology co. LTD,

5. Your ethics statement must appear in the Methods section of your manuscript. If your ethics statement is written in any section besides the Methods, please move it to the Methods section and delete it from any other section. Please also ensure that your ethics statement is included in your manuscript, as the ethics section of your online submission will not be published alongside your manuscript.

6. Please include captions for your Supporting Information files at the end of your manuscript, and update any in-text citations to match accordingly. Please see our Supporting Information guidelines for more information: http://journals.plos.org/plosone/s/supporting-information

**Comments to the Author**

1. Is the manuscript technically sound, and do the data support the conclusions?

Reviewer #1: Yes

Reviewer #2: Yes

2. Has the statistical analysis been performed appropriately and rigorously? 

Reviewer #1: No

Reviewer #2: Yes

3. Have the authors made all data underlying the findings in their manuscript fully available?

Reviewer #1: Yes

Reviewer #2: Yes

4. Is the manuscript presented in an intelligible fashion and written in standard English?

Reviewer #1: Yes

Reviewer #2: Yes

5. Review Comments to the Author

Reviewer #1:

-Line 287 the authors mentioned w1-w11 but the formula only show up to w9, where are w10 and w11?

-Please explain how alpha and beta values are determined

-The english need to be improved.

-Include a statistical test to show significancy between the different algorithms that were compared.

-How were defined the number of samples selected as training and test sets? are these amounts enough? Why didn't the authors use cross validation? Where was the model implemented and tested (computer characteristics, software, etc.)?

Reviewer #2:

1. Background missing. Clarity of presentation is needs to improved via clearly specifying objectives, focus and motivation of the proposed work

2. Recent Research: More recent references need to be added and discussed in the literature section and background information about existing approaches in the context of proposed work addressed in this paper:

https://www.sciencedirect.com/science/article/abs/pii/S1568494618306422

https://link.springer.com/article/10.1007%2Fs00500-018-3177-y

https://www.sciencedirect.com/science/article/abs/pii/S1568494615001209

https://link.springer.com/article/10.1007/s11227-018-2279-8

3.Open issues, limitations of the existing approaches or prevalent trend /issues of Predicting human body composition using a modified adaptive genetic algorithm can be elucidated.

4. Add the contents in the abstract of the paper and highlight the impact of the proposed work.

5. The method/approach in the context of proposed work should be write in detail.

6. Real time case studies might be add more importance of this chapter. The authors should add the case study and its results.

7.Result and discussion should be rewritten to summarize the findings/significance of the work. Please discuss the research limitations and future works in the conclusion.

8. To explore Comparative results with of existing approaches/methods with relate to the proposed work.

---

## [Author Response · Author response to Decision Letter 0]

28 Jan 2020

Original Manuscript ID: PONE-D-19-20446 

Original Article Title: “Predicting human body composition using a modified adaptive genetic algorithm ”

To: PLOS ONE Editor

Re: Response to reviewers

Dear Editor,

Thank you for allowing a resubmission of our manuscript, with an opportunity to address the reviewers’ comments.

We are uploading (a) our point-by-point response to the comments (below) (response to reviewers), (b) an updated manuscript with yellow highlighting indicating changes, and (c) a clean updated manuscript without highlights. In addition, we proofread the previous simulation results and updated the data in Table 5 and the ethics statement was added to the Method section of the manuscript.

We tried our best to improve the manuscript and made some changes in the manuscript.  These changes will not influence the content and framework of the paper. We appreciate for Editors/Reviewers’ warm work earnestly, and hope that the correction will meet with approval.

Best regards,

<Bo Chen> et al.

---

## [Decision Letter · Decision Letter 1]

5 Mar 2020

PONE-D-19-20446R1

Predicting human body composition using a modified adaptive genetic algorithm

PLOS ONE

Dear Mr. Chen,

Thank you for submitting your manuscript to PLOS ONE. After careful consideration, we feel that it has merit but does not fully meet PLOS ONE’s publication criteria as it currently stands. Therefore, we invite you to submit a revised version of the manuscript that addresses the points raised during the review process.

We would appreciate receiving your revised manuscript by Apr 19 2020 11:59PM. To enhance the reproducibility of your results, we recommend that if applicable you deposit your laboratory protocols in protocols.io, where a protocol can be assigned its own identifier (DOI) such that it can be cited independently in the future. For instructions see: http://journals.plos.org/plosone/s/submission-guidelines#loc-laboratory-protocols

We look forward to receiving your revised manuscript.

Kind regards,

Le Hoang Son, Ph.D

Academic Editor

PLOS ONE

**Comments to the Author**

1. If the authors have adequately addressed your comments raised in a previous round of review and you feel that this manuscript is now acceptable for publication, you may indicate that here to bypass the “Comments to the Author” section, enter your conflict of interest statement in the “Confidential to Editor” section, and submit your "Accept" recommendation.

Reviewer #1: All comments have been addressed

Reviewer #2: All comments have been addressed

Reviewer #3: All comments have been addressed

2. Is the manuscript technically sound, and do the data support the conclusions?

Reviewer #1: Yes

Reviewer #2: Yes

Reviewer #3: Partly

3. Has the statistical analysis been performed appropriately and rigorously? 

Reviewer #1: No

Reviewer #2: Yes

Reviewer #3: No

4. Have the authors made all data underlying the findings in their manuscript fully available?

Reviewer #1: Yes

Reviewer #2: Yes

Reviewer #3: No

5. Is the manuscript presented in an intelligible fashion and written in standard English?

Reviewer #1: Yes

Reviewer #2: Yes

Reviewer #3: Yes

6. Review Comments to the Author

**Reviewer #1**: (No Response)

**Reviewer #2**: (No Response)

**Reviewer #3**:

1.Why the author has chosen the modified adaptive genetic algorithm Predicting 1 human body composition ?.As there are several algorithms available that can predict better results.

2.Needs more explanation of Table.1 as well as Table.2 parameters in the result and discussion section.

3.In Table.4 the authors present theComparison of the training time for different models but how actually derived the Maximum fitness degree and Training time(s) values are unclear.

Elaborate with suitable examples. Data seems to fabricate.

4.Very hard to find the novelty of this manuscript? The authors suggested highlighting their own contributions.

5.If possible explain briefly the dataset and parameters which have been used in this article.

---

## [Author Response · Author response to Decision Letter 1]

18 Mar 2020

To: PLOS ONE Editor

Re: Response to reviewers

Dear Editor,

Thank you for allowing a resubmission of our manuscript, with an opportunity to address the reviewers’ comments.

We are uploading (a) our point-by-point response to the comments (response to reviewers), (b) an updated manuscript with yellow highlighting indicating changes, and (c) a clean updated manuscript without highlights.

We tried our best to improve the manuscript and made some changes in the manuscript. These changes will not influence the content and framework of the paper. We appreciate for Editors/Reviewers’ warm work earnestly, and hope that the correction will meet with approval.

Best regards,

<Bo Chen> et al.

---

## [Decision Letter · Decision Letter 2]

29 Apr 2020

PONE-D-19-20446R2

Predicting human body composition using a modified adaptive genetic algorithm

PLOS ONE

Dear Mr. Chen,

Thank you for submitting your manuscript to PLOS ONE. After careful consideration, we feel that it has merit but does not fully meet PLOS ONE’s publication criteria as it currently stands. Therefore, we invite you to submit a revised version of the manuscript that addresses the points raised during the review process.

We would appreciate receiving your revised manuscript by Jun 13 2020 11:59PM. To enhance the reproducibility of your results, we recommend that if applicable you deposit your laboratory protocols in protocols.io, where a protocol can be assigned its own identifier (DOI) such that it can be cited independently in the future. For instructions see: http://journals.plos.org/plosone/s/submission-guidelines#loc-laboratory-protocols

We look forward to receiving your revised manuscript.

Kind regards,

Le Hoang Son, Ph.D

Academic Editor

PLOS ONE

**Comments to the Author**

1. If the authors have adequately addressed your comments raised in a previous round of review and you feel that this manuscript is now acceptable for publication, you may indicate that here to bypass the “Comments to the Author” section, enter your conflict of interest statement in the “Confidential to Editor” section, and submit your "Accept" recommendation.

Reviewer #1: All comments have been addressed

Reviewer #2: All comments have been addressed

Reviewer #3: All comments have been addressed

2. Is the manuscript technically sound, and do the data support the conclusions?

Reviewer #1: Yes

Reviewer #2: Yes

Reviewer #3: Partly

3. Has the statistical analysis been performed appropriately and rigorously? 

Reviewer #1: Yes

Reviewer #2: Yes

Reviewer #3: Yes

4. Have the authors made all data underlying the findings in their manuscript fully available?

Reviewer #1: Yes

Reviewer #2: Yes

Reviewer #3: Yes

5. Is the manuscript presented in an intelligible fashion and written in standard English?

Reviewer #1: Yes

Reviewer #2: Yes

Reviewer #3: No

6. Review Comments to the Author

**Editor: **We appreciate that the revised paper has been significantly improved, but still some problems remained:

1> The authors should ask a native English speaker or consult a professional English editting service to proofread the paper. Please upload the certificate of language correction in the submission. Some examples, to name a few:

-- Abstract: "Background: Changes to human body composition reflect, to some extent, changes in health status"  (Incomplete sentence).

-- Introduction: "However, once the regression equation is determined, 68 it becomes difficult to adjust it according to the situation, meaning adaptability is poor." (Incomplete sentence).

-- Improved RReliefF algorithm: "Where ik x is the k-th body physiological parameter value of sample "   (where instead of Where).

..

2) Equations need re-typed. The current version lacks of clear formulae and equations. All equations must be numberred.

3) Please append statistical analysis for the experimental results in Tables 4-5 to prove the achieved results being statistically significance. 

4) Future works must be appended by adding 3-5 more solid works from this sentence: "In future research work, we will consider how to more effectively reduce the time complexity of the algorithm and cross-validation will be used to test model performance in future research."

5) The coherence to PLOS ONE should be explained clearly in the Introduction and through recent related articles.

6) The authors should consider to change the paper title to reflect more contribution and coherence to PLOS. 

"Predicting human body composition using a modified adaptive genetic algorithm"

What is modified here?

**Reviewer #1**: All the reviewers' observations were addressed. This last version of the manuscript shows a remarkable improvement compared to its initial version.

**Reviewer #2**: All the comments are properly addressed

No further review is required.

The manuscript ready to be accepted

**Reviewer #3**:

1. Author should highlight the contribution

2. More emphasize on the dataset.Sources of dataset should required to mention

3. More explanations on the Figures and Tables.

---

## [Author Response · Author response to Decision Letter 2]

17 May 2020

Manuscript Number: PONE-D-19-20446R2 

Article Title: “Predicting human body composition using a modified adaptive genetic algorithm with a novel selection operator”

To: PLOS ONE Editor

Re: Response to reviewers

Dear Editor,

Thank you for allowing a resubmission of our manuscript with the opportunity to address the reviewers’ comments.

We are uploading: (a) our point-by-point response to the comments from the reviewers and editor (below); (b) an updated manuscript, where yellow highlighting indicating changes; and (c) a clean, updated manuscript without highlights.

We have tried our best to improve the manuscript. The changes we have made will not influence the content or framework of the paper. We sincerely appreciate the work of the editor and reviewers and hope that our corrections will meet with approval.

Best regards,

<Bo Chen> et al.

---

## [Decision Letter · Decision Letter 3]

23 Jun 2020

Predicting human body composition using a modified adaptive genetic algorithm with a novel selection operator

PONE-D-19-20446R3

Dear Dr. Chen,

We’re pleased to inform you that your manuscript has been judged scientifically suitable for publication and will be formally accepted for publication once it meets all outstanding technical requirements.

Kind regards,

**Le Hoang Son, Ph.D**

Academic Editor

PLOS ONE

**Comments to the Author**

1. If the authors have adequately addressed your comments raised in a previous round of review and you feel that this manuscript is now acceptable for publication, you may indicate that here to bypass the “Comments to the Author” section, enter your conflict of interest statement in the “Confidential to Editor” section, and submit your "Accept" recommendation.

Reviewer #2: All comments have been addressed

Reviewer #3: All comments have been addressed

2. Is the manuscript technically sound, and do the data support the conclusions?

Reviewer #2: Yes

Reviewer #3: Yes

3. Has the statistical analysis been performed appropriately and rigorously? 

Reviewer #2: Yes

Reviewer #3: Yes

4. Have the authors made all data underlying the findings in their manuscript fully available?

Reviewer #2: Yes

Reviewer #3: Yes

5. Is the manuscript presented in an intelligible fashion and written in standard English?

Reviewer #2: Yes

Reviewer #3: (No Response)

6. Review Comments to the Author

**Reviewer #2**: Article is interesting and address the current research problem. All the comments are addressed. The organization of the paper is very good with different sections and paragraphs are clear, which makes the paper easy to follow. I encourage the authors to think more on the overall structure of the manuscript.

**Reviewer #3**: (No Response)

---

## [Editor Report · Acceptance letter]

1 Jul 2020

PONE-D-19-20446R3 

Predicting human body composition using a modified adaptive genetic algorithm with a novel selection operator 

Dear Dr. Chen:

I'm pleased to inform you that your manuscript has been deemed suitable for publication in PLOS ONE. Congratulations! Your manuscript is now with our production department. 

Kind regards, 

on behalf of

Prof. Le Hoang Son 

Academic Editor

PLOS ONE